# Neighborhood Disadvantage and Tobacco Retail Outlet and Vape Shop Outlet Rates

**DOI:** 10.3390/ijerph17082864

**Published:** 2020-04-21

**Authors:** David C. Wheeler, Elizabeth K. Do, Rashelle B. Hayes, Kendall Fugate-Laus, Westley L. Fallavollita, Colleen Hughes, Bernard F. Fuemmeler

**Affiliations:** 1Department of Biostatistics, Virginia Commonwealth University, Richmond, VA 23298, USA; 2Massey Cancer Center, Virginia Commonwealth University, Richmond, VA 23298, USA; elizabeth.do@vcuhealth.org (E.K.D.); rashelle.hayes@vcuhealth.org (R.B.H.); bernard.fuemmeler@vcuhealth.org (B.F.F.); 3Department of Health Behavior & Policy, Virginia Commonwealth University, Richmond, VA 23298, USA; kcfugatelaus@vcu.edu (K.F.-L.); fallavolliwl@mymail.vcu.edu (W.L.F.); 4Department of Psychiatry, Virginia Commonwealth University, Richmond, VA 23298, USA; 5Department of Behavioral Health & Developmental Services, Richmond, VA 23298, USA; colleen.hughes@dbhds.virginia.gov

**Keywords:** density, geospatial, neighborhood disadvantage, outlet, retail, SES, tobacco vape

## Abstract

Neighborhood-level socioeconomic variables, such as the proportion of minority and low-income residents, have been associated with a greater density of tobacco retail outlets (TROs), though less is known about the degree to which these neighborhood indicators are related to vape shop outlet (VSO) density. Many studies of TROs and neighborhood characteristics include only a small set of variables and also fail to take into account the correlation among these variables. Using a carefully curated database of all TROs and VSOs in Virginia (2016–2018), we developed a Bayesian model to estimate a neighborhood disadvantage index and examine its association with rates of outlets across census tracts while also accounting for correlations among variables. Models included 12 census tract variables from the American Community Survey. Results showed that increasing neighborhood disadvantage was associated with a 63% and 64% increase in TRO and VSO risk, respectively. Important variables associated with TRO rates included % renter occupied housing, inverse median gross rent, inverse median monthly housing costs, inverse median monthly housing costs, and % vacant housing units. Important variables associated with VSO rates were % renter occupied housing and % Hispanic population. There were several spatial clusters of significantly elevated risk for TROs and VSOs in western and eastern Virginia.

## 1. Introduction

Tobacco use is the leading cause of preventable death in the United States, and disproportionately affects individuals from racial and ethnic minority populations [1] and individuals from lower socioeconomic status (SES), including those with lower educational and occupational attainment [2,3]. Existing disparities in tobacco use and its health-related outcomes have been attributed to a high tobacco retail outlet (TRO) density in disadvantaged neighborhoods, where such outlets increase the visibility of tobacco marketing, make tobacco products more available, and create a normative culture accepting of tobacco use [4]. TRO placement is likely not random. A better understanding of where TROs are statistically more likely to be located accounting for the population density within a state or region could help identify “hot-spots” that could be targeted for community outreach and education. Likewise, knowing if there are particular neighborhood characteristics that favor a higher density of TROs would inform policies that seek to limit the density of TROs in these “high-risk” neighborhoods.

### 1.1. Neighborhood Characteristics and Tobacco Retail Outlet Density

A number of studies have attempted to link neighborhood characteristics with TRO density with varying results. For instance, a study conducted in New York found associations between TRO density and low-income high-minority populations; however, associations were not consistent across all regions of the state—some regions showed no associations whereas others showed significant and positive associations [5]. Another study conducted in Boston, Massachusetts found no association in a multivariate model between demographic characteristics (i.e., % non-Hispanic black, Hispanic, families in poverty) and TRO density [1]. In another study of socioeconomic status and TRO density in Maryland, the investigators found that higher socioeconomic jurisdictions had a lower tobacco outlet density than lower socioeconomic jurisdictions, despite the populations being similar with respect to race (i.e., predominately white) [6].

### 1.2. Neighborhood Characteristics and Vape Shop Outlet Density

While recent work around understanding how TROs are located with respect to neighborhood variables has been expanding, there has been an exponential increase in electronic cigarette (e-cigarette) use. These tobacco products are sometimes sold in the same place as traditional tobacco products, but they are also sold in specialty shops (i.e., vape shop outlets (VSOs)). To date, there have been limited studies examining VSO density in relation to neighborhood variables; however, similar to the TRO literature, VSO density in relation to neighborhood variables has been mixed. At least one study has shown that these retail outlets are more likely to be located in areas with predominately non-Hispanic white residents [7], whereas others have found that VSOs are concentrated in locations with higher levels of sociodemographic disparities, such as concentrated poverty and racial segregation [8,9].

### 1.3. Statistical Approaches for Examining Associations between Neighborhood Characteristics and TRO/VSO Density

The lack of consistent findings across studies suggest possible methodological differences across these studies. Many extant studies have utilized standard statistical approaches, such as one-way analyses of variance (ANOVA), to compare predictors and criterion variables [10,11] and linear regression analyses [12] to establish a statistical relationship between the density of tobacco retail outlets and neighborhood variables (e.g., % black population, % Hispanic population, median household income, etc.). By taking this approach, the correlation of observations over geographical space is ignored, potentially leading to biased effect estimates and inference regarding the associations between TRO and VSO density and social disadvantage. Improved methodologies for understanding these associations between neighborhood geospatial variables and TROs/VSOs could help bring greater clarity to the ways in which TRO/VSO density varies in relation to neighborhood variables.

To overcome the limitation noted above, different spatial regression techniques have been developed to account for geographic clustering in the data [13,14]. These spatial regression techniques include calculation of the spatial autocorrelation in the residuals resulting from autocorrelation of the dependent variable (e.g., spatial lag), spatial autocorrelation in the error term due to spatially autocorrelated predictors not included in the models (e.g., spatial error), and conducting Lagrange multiplier tests to determine whether a spatial model or standard ordinary least squares modeling approach was most appropriate [5]. Although these approaches can account for spatial dependence in the data, they do not estimate an index of correlated components, such as what is inherent with respect to neighborhood variables, where, for example, a higher % of racial/ethnic minorities is highly correlated with other indicators of neighborhood disadvantage.

### 1.4. Research Objectives

The objective of this study was to identify significant clusters and hotspots of vape and tobacco retailer density while finding important neighborhood socioeconomic status (SES) variables associated with outlet rates. To accomplish this objective, we developed Bayesian hierarchical models for TRO and VSO rates at the census tract level that handled spatial autocorrelation and correlated neighborhood variables in a neighborhood disadvantage index (NDI). We hypothesized that TRO rates would increase with increases in neighborhood social disadvantage (as measured by the NDI). Given previous findings, we also hypothesized that the variables measuring household income and race/ethnicity would have the largest influence on this association and that important variables in the NDI could be different for TRO rates and VSO rates.

### 1.5. Theoretical Framework

Persistent and growing disparities in tobacco use and tobacco-related health outcomes reflect larger structural forces shaping the social context of everyday life [15]. The social-ecological model provides a useful framework for examining the ways in which social-contextual factors (such as race/ethnicity, income, housing) help to explain tobacco use, while also identifying areas of potential intervention at the individual, family/peer, community, and societal level [16]. For example, smoking is associated with low income and often clusters with other social-contextual factors, such as unemployment, lack of social support, and having unmet needs regarding education, safety, food, and medical care [17,18,19]. By identifying significant clusters and hotspots of vape and tobacco retailer density while finding important neighborhood SES variables that explain variation in TRO and VSO rates, we are able to identify communities that require intervention, while also determining patterns of social circumstance that occur with neighborhood social disadvantage that can be used to inform and improve future prevention and intervention efforts.

## 2. Materials and Methods

### 2.1. Data Sources and Measures

This study utilized publicly available data on tobacco and vape retail outlet listings in Virginia and US Census data from the American Community Survey (ACS) to estimate and explain variation in TRO and VSO rates at the census tract level. VA zip code is an important state for the study of tobacco use. Within VA zip code, there are no licensing requirements for the sale of tobacco and no existing regulations that affect where tobacco and nicotine containing products are sold. VA zip code also has one of the lowest excise taxes on both cigarette and non-cigarette products in the United States [20]. Additionally, although state rates for smoking are generally lower when compared to the national average [21], tobacco use remains the leading preventable cause of death within the state. Further, state-level data has demonstrated that tobacco use differs by geographic region, race/ethnicity, income, and education across the state [22], suggesting that these are important variables to investigate in our analyses. Study protocols were approved by the Institutional Review Board at Virginia Commonwealth University (HM20013609).

### 2.2. Tobacco and Vape Retail Outlet Listings

Between 2016 and 2018, the Department of Behavioral Health and Developmental Services within the Virginia Department of Health (VDH) used a standardized methodology developed by CounterTools to curate a database of all TROs/VSOs in Virginia. This involved using publicly available data sources and a team of 40 community board partners to “ground truth” every TRO/VSO within Virginia by driving every primary and secondary road across the state to verify retail addresses or add those that were found that were not in the database. From VDH, we obtained TRO listings for N = 5609 tobacco retail outlets in Virginia in December 2018. Store type (e.g., convenience stores and gas stations, grocery stores, mass merchandisers, drug stores or pharmacies, tobacco shops, e-cigarette and vape shops, bars and restaurants, and hookah lounges) was indicated for most TRO/VSOs. For retailers that were missing store type information (n = 461), our research staff used Google Maps, Yelp, Yellow Pages, and business websites to confirm the store type for 452 of the 461 missing this information. After validation, we calculated and used data from the 5600 TROs and 167 VSOs in Virginia in December 2018.

### 2.3. TRO and VSO Rates

We assigned each TRO and VSO to a census tract based upon longitude and latitude coordinates provided by CounterTools. We then calculated the rates of TROs and VSOs per total number of households within a census tract as the outcome variables.

### 2.4. American Community Survey Data

The ACS is administered annually to three million households, representative of the US population. Participants complete a questionnaire and report their household’s social and economic information. We used five-year (2012–2016) ACS estimates of 12 variables at the census tract level to construct neighborhood deprivation indices. These variables were: Gini index of income inequality, % black population, % with bachelor’s degree, % families in poverty, % households with public assistance income, % vacant housing units, % renter occupied housing units, median household income, median gross rent, median monthly housing costs, % Hispanic population, and % US citizen. We have used similar variables to estimate neighborhood disadvantage indices previously [15,16]. Among the 12 SES variables, there were several with missing values. After excluding 87 census tracts with missing values for at least one variable, there were 1820 census tracts for modeling TRO and VSO rates.

### 2.5. Statistical Analysis

We used Bayesian regression models for the TRO rate and the VSO rate separately, assuming that the TRO and VSO count in each census tract was $y_{i}∼Poisson\left(\theta_{i}E_{i}\right)$ with a relative risk $\theta_{i}$ and expected count $E_{i}$. The expected count for each census tract was calculated as the product of the overall TRO or VSO rate $r=\left(\overset{n}{\underset{i=r}{\sum }}y_{i}/\overset{n}{\underset{i=1}{\sum }}h_{i}\right)$ in the state and the number of households $h_{i}$ in the tract. We used four models of increasing complexity as candidates for explaining the variation in TRO/VSO rates with a neighborhood deprivation index. We considered the following candidates for modeling the log of the relative risk of TROs or VSOs:(1)$$\log(\theta_{i})=\beta_{0}+\beta_{1}\left(\sum_{j=1}^{C}w_{j}q_{ij}\right)+u_{i},$$
(2)$$\log(\theta_{i})=\beta_{0}+\beta_{1}\left(\sum_{j=1}^{C}w_{j}q_{ij}\right)+v_{i},$$
(3)$$\log(\theta_{i})=\beta_{0}+\beta_{1}\left(\sum_{j=1}^{C}w_{j}q_{ij}\right)+u_{i}+v_{i},$$
(4)$$\log(\theta_{i})=\beta_{0}+\beta_{1}\left(\sum_{j=1}^{C}w_{j}q_{ij}\right)+\alpha_{i}u_{i}+\left(1-\alpha_{i}\right)v_{i},$$
where $\beta_{0}$ is the intercept, $\beta_{1}$ is the effect for the neighborhood disadvantage index, $u_{i}$ and $v_{i}$ are tract level random effects, and $\alpha_{i}$ is a mixing parameter.

The first model is the base index model that includes unstructured tract-level random effects, the second model includes spatially structured tract-level random effects, the third model includes both unstructured and spatially structured tract-level random effects (convolution model), and the fourth is a convolution mixture model with a mixing parameter on the unstructured random effect and the spatially structured random effect. In model 1, the heterogeneity in TRO/VSO rates not explained by the deprivation index is assumed to be random over space, while in model 2 it is assumed to be spatially correlated. In model 3, it can be both spatially correlated and random over space. In model 4, the mixing parameter is estimated to allow the data to inform on the nature of the heterogeneity in the rates. The uncorrelated random effects model (model 1) is used as a comparison to explore the assumption of spatial dependence in the data. The choice of the convolution model (model 3) is motivated by a possible spatial correlation in the rates. Model 4 is included to allow the influence of the unstructured and spatially structured random effects to fluctuate through the addition of the mixing parameter. Within these Bayesian models, population density (e.g., with variation across urban, suburban, and rural areas) is directly considered in the TRO/VSO rate and also accounted for through SES covariate patterns that vary over space in relation to population density. However, proximity between outlets within a census tract (e.g., the average distance between retailers) is not considered in these models.

We specified the neighborhood disadvantage index for each tract using a weighted combination $\overset{C}{\underset{j=1}{\sum }}w_{j}q_{j}$ of the quantiles $q_{1},\ldots ,q_{c}$ of the SES variables $x_{1},\ldots ,x_{c}$, where the weights $w_{1},\ldots ,w_{C}$ were estimated in the model. The weight $w_{j}$ represents the relative importance of the $j_{th}$ SES variable in the index. We used deciles of the SES variables to account for different scales of the variables, limit the effect of outliers, de-correlate the variables, and acknowledge uncertainty in the ACS covariates. We used C=12 SES variables in the index. The SES variables were defined to reflect a hypothesized positive association of the index with TRO rates. Some of the ACS variables were redefined to have a positive association with TRO counts in univariate analyses. These variables were median household income, median gross rent, median monthly housing costs, and % with a bachelor’s degree. We inverted the income, rent, and housing cost variables by using the formula $\max\left(x\right)-x_{i}$, where $x_{i}$ is the value of the variable. We redefined education to be % without a bachelor’s degree using the formula $1-x_{i}$.

The Bayesian model specification is completed with the definition of prior distributions for the priors. The assumption of spatial correlation in the rates was implemented through an intrinsic conditional autoregressive (ICAR) prior [17], where each random effect has the conditional distribution given by $v_{i}|v_{-i}∼Normal\left({\bar{v}}_{l},1/\tau_{v}\delta_{i}\right)$ with ${\bar{v}}_{l}=\underset{j\;in\;\omega_{i}}{\sum }v_{j}/\delta_{i}$, where $\delta_{i}$ represents the number of neighbors in set $\omega_{i}$ and precision $\tau_{v}=1/\sigma_{v}^{2}$ and $\sigma_{v}∼Uniform\left(0,100\right)$. We defined spatial structure using binary neighborhood weighting and queen contiguity. The index weights were given a Dirichlet prior with parameters $\alpha =\left(\alpha_{1},\ldots ,\alpha_{C}\right)$. The Dirichlet prior was used because it assures that the SES variable weights $w_{j}\in \left(0,1\right)$ and $\overset{C}{\underset{j=1}{\sum }}w_{j}=1$. The prior for the unstructured random effects was $u_{i}∼Normal\left(0,\tau_{u}\right)$ with precision $\tau_{u}=1/\sigma_{u}^{2}$ and $\sigma_{u}∼Uniform\left(0,100\right)$. The mixing parameter $\alpha_{i}$ followed a Beta(1,1) prior. The intercept followed an improper uniform distribution α∼dflat(), while the index regression coefficient had a vague normal prior $\beta_{1}∼Normal\left(1,\tau_{1}\right)$ with precision $\tau_{1}=1/\sigma_{1}^{2}$ and $\sigma_{1}∼Uniform\left(0,100\right)$.

We used Markov Chain Monte Carlo (MCMC) to estimate the model parameters with a total of 60,000 iterations from one chain and a thinning parameter of one, where the first 30,000 iterations were used for burn-in. We assessed the convergence of the MCMC algorithm for parameters of interest using the Geweke convergence diagnostic [23]. A parameter was considered to have converged if its diagnostic absolute value was less than 2. Among the four candidate models, the one with the lowest deviance information criterion (DIC) measure of goodness-of-fit was chosen as the best model [24]. The 95% credible interval for the relative risk was used to determine the statistical significance of the disadvantage index; it was deemed statistically significant if the interval did not contain the value of 1. We fit the Bayesian models using WinBUGS1.4.1 [24] and completed all other analyses in the R computing environment.

We identified census tracts as being significantly elevated for TRO/VSO risk using posterior estimates of exceedance probabilities for the relative risk defined as $q_{i}^{c}=\overset{m+G}{\underset{g=m+1}{\sum }}\frac{I(\theta_{i}>c)}{G}$, where *m* represents the burn-in (30,000 iterations) and *G* represents the number of posterior samples after the burn-in (60,000 iterations) [19]. The relative risk *c = 1* was used as a threshold value for $\theta_{i}$. Census tracts with an exceedance probability greater than 0.90 were deemed to have significant elevated risk of TROs/VSOs.

## 3. Results

There were 1480 census tracts with at least one TRO and 427 with no TROs. There were 137 tracts with at least one VSO and 1770 tracts with no VSOs. Retail outlets consisted of convenience stores and gas stations (59.1%), grocery stores (14.0%), mass merchandisers (9.4%), drug stores or pharmacies (5.8%), tobacco shops (4.2%), e-cigarette and vape shops (3.0%), store type either unlisted or unknown (2.6%), bars and restaurants (1.0%), and hookah lounges (0.6%). Within this specific dataset, VSOs were a subset of TROs.

A steady decrease in model DIC values show that there was a consistent improvement in the goodness-of-fit going from model 1 to model 4 (Table 1). A decrease in 10 or more in the DIC indicates a meaningful improvement in the fit. There was a dramatic decrease in the goodness-of-fit going from model 1 to model 2, indicating the presence of significant spatial correlation in the TRO/VSO rates. A Moran’s I test also indicated significant spatial autocorrelation (*p* = 0.001) in the rates for TROs and marginal significance for VSOs (*p* = 0.10). Adding independent random effects to the model with spatially correlated random effects (model 3) resulted in a smaller yet meaningful decrease in DIC. Including a mixing parameter to the convolution model (model 4) resulted in a large decrease in DIC, revealing that the flexibility in modeling residual risk provided by the convolution mixture model was beneficial. While the effective number of parameters increased substantially from model 1 to model 4 according to the pD statistic, the decrease in deviance much exceeded the increase in model complexity, resulting in an improved fit. Hence, the most complex model (model 4) had the best goodness-of-fit and is used for explaining variation in TRO/VSO rates for the remainder of the results.

The relative risk estimated by the Bayesian model for TROs is mapped in Figure 1 and for VSOs in Figure 2. The relative risk of 1 is used as a reference in the color ramp; blues are below average risk and reds are above average risk. The pattern of relative risk for TROs shows an overall reduced risk in northern Virginia and the capital city of Richmond, and elevated risk in the Eastern Shore. Some census tracts on the southern border with North Carolina or western border with West Virginia also have elevated risk. For VSOs, there is pronounced elevated risk in Norfolk Virginia Beach Charlottesville, Harrisonburg, and parts of rural southwestern Virginia. Northeastern Virginia on the border of Washington, DC and areas surrounding Richmond have a reduced risk for VSOs. Overall, the pattern of risk for VSOs and TROs differs. The areas of significantly elevated risk according to the posterior exceedance probabilities are mapped in Figure 3 for TROs and Figure 4 for VSOs. Every shaded census tract is significantly elevated in relative risk. There were 238 significant tracts for TROs and 25 for VSOs. These significant census tracts are listed in the Appendix A. There is a cluster of significantly elevated risk areas in the Eastern Shore for TROs as well as several smaller clusters in the southern and western portion of the state. For VSOs, there is significantly elevated risk in the Norfolk and Virginia Beach area and several “hotspots” in northern and western Virginia.

The estimated NDI for all TROs was significantly and positively associated with TRO density with a relative risk (RR) = 1.22 (CI: 1.19, 1.27), meaning a one-unit increase in the NDI was associated with a 22% increase in risk of TROs. In interpreting the index weights, the estimated weight per variable would be 0.083 (or 1/12) if each variable contributed equally to the association. Where variables have estimated weights of >0.083, it suggests that the variables explain more of the variation in the association between NDI and TRO and VSO density. Therefore, we used this threshold for identifying important variables when interpreting the indices. The most important variables in the estimated NDI for all TROs, according to the estimated index weights (Table 2) were % renter occupied housing units (0.23), inverse median gross rent (0.20), % without bachelor’s degree (0.10), inverse median monthly housing costs (0.09), and % vacant housing units (0.09).

The estimated NDI for VSOs was significantly and positively associated with VSO density, with an RR = 1.20 (CI: 1.05, 1.38). In this index, the most important variable was % renter occupied housing units with an index weight of 0.46. The other important variable was % Hispanic population with a weight of 0.20. No other variables in the index had weights above the equal-weight threshold of 0.083.

## 4. Discussion

In this study, we developed a novel Bayesian model to explain variation in TRO and VSO rates using specific neighborhood-level SES variables, while accounting for the correlated nature of the SES variables and the correlation of outlet rates over space. Results from this study show geographic variability in the placement of TRO and VSOs (Figure 1 and Figure 2), with statistically elevated risk areas (spatial clusters and hotspots) in the eastern portions, south central, and western Appalachian part of the state (Figure 3 and Figure 4). Using this method, we found that greater neighborhood disadvantage, as measured by 12 neighborhood-level variables, was significantly associated with increased risk of TRO and VSO density across census tracts in Virginia.

However, not all variables in the index were equally important in explaining the association between neighborhood disadvantage and TRO or VSO density. Variables reflecting neighborhood socioeconomic disadvantage, such as lower median household income, higher % renter occupied housing units, lower gross rent, and lower median monthly housing, were more important for TRO density, whereas other variables, such as % black, were less important. With respect to VSOs, a higher % renter occupied housing was strongly associated with an increased VSO density and made up the majority of the index. Our finding of an association with % of renters may be related to e-cigarette product use targeting among youth and young adults: In addition to being more likely to rent their homes [25], young adults are also more likely to be exposed to and use e-cigarette products [26].

The second most important variable for VSO density was % Hispanic population. This differed from our expectation that VSO rates should be more likely in areas with higher % white population based on findings from the existing literature [7]. A possible explanation for why we found an association between VSO density and % Hispanic population could be related to the growing Hispanic and Latino population within Virginia, as well as the increasing prevalence of tobacco use within this population. Within the US, the Hispanic and Latino population is the largest racial/ethnic minority group. This population has also become a rapidly growing market for tobacco advertisements and promotion for the tobacco industry [27,28]. Prior research suggests that tobacco use among Hispanic adolescents has been increasing, such that prevalence rates of tobacco and e-cigarette use among Hispanic adolescents are approaching prevalence rates found among non-Hispanic black and white populations [29,30,31]. These trends are also found in Virginia, where the Hispanic and Latino population has grown from 4.7% to 9.3% from 2000 to 2017 [32] and the estimated prevalence of ever electronic nicotine device use, including e-cigarettes, within the Hispanic and Latinx population is 24.5%. This estimated prevalence is greater than those of all other race/ethnicity groups (e.g., 18.8% of non-Hispanic whites, 15.7% of non-Hispanic blacks, 21.0% of Asians, and 20.0% of individuals who report other race/ethnicities) [22].

What our results and those reported from other geospatial studies conducted in New Jersey [13] and Boston, Massachusetts [1] suggest is that income, race, and ethnicity are important variables in determining TRO and VSO placement. What differs across these studies are conclusions regarding the relative importance of these correlated variables in determining TRO and VSO placement. In our study, increased % Hispanic population and increased % black population were associated with increased TRO and VSO rates. However, race and ethnicity variables received far less weight than socioeconomic variables in the indices. Our results differ from a study conducted in New Jersey using a spatial lag model that found % Hispanic population to be the dominant demographic factor associated with TRO placement, followed by median household income and % black population [13]. The study conducted in Boston found a positive association between % Hispanic population and TRO density, which became statistically non-significant in multivariate spatial lag models that also accounted for % black population and % families in poverty [1]. Inconsistencies in the relative importance of these variables in explaining TRO and VSO density may reflect dissimilar state and local tobacco control policies, as well as differences in the statistical approach. More progressive locales may be better able to implement policies that impact TRO and VSO placement, growth, and/or closure (e.g., excise taxes, prohibitory regulations, such as restrictions on placement within certain distances of specific locations). Thus, continued characterization of TRO and VSO placement is necessary for understanding how correlated variables are associated with TRO and VSO density over time and across states or regions.

Our study meets this need by being the only study that has characterized TRO and VSO placement utilizing a Bayesian hierarchical modeling approach for estimating a neighborhood disadvantage index to explain variation in TRO and VSO density. It is beneficial to estimate the index weights from the data through the Bayesian modeling approach, instead of assigning a priori weights to potential risk factors, such as with summed z-score index approaches whose measures are not likely to capture the complexity of SES across all geographic areas. Contrary to the common equal weighting assumption in the summed z-score approach, we found strong deviations in the weighting of the SES variables in our estimated neighborhood disadvantage index. Moreover, the Bayesian framework flexibly allows a model specification that includes residual confounding that is a mixture of spatially structured and unstructured random effects. Even in the complex mixture model of spatially structured and unstructured random effects, the neighborhood disadvantage index was significantly associated with both TRO and VSO risk. Another benefit of the Bayesian modeling approach is the ability to easily identify areas of significantly elevated risk using exceedance probabilities, as demonstrated by the identification of many census tracts of significantly elevated risk for TROs and VSOs in the southeastern, south central, and western portions of the state. Identifying specific regional or city differences in elevated risk for TRO or VSO placement may help in implementing future local policy or prevention marketing strategies. Given that TRO and VSO placement is associated with an index of neighborhood social disadvantage, future policies need to take into consideration how TRO and VSO placement could be associated with existing tobacco use disparities associated with social, economic, and demographic factors.

Although there are many strengths to this study, there are also a few limitations that should be considered. First, this study used data from one state, which may or may not generalize to others with a different mixture of social, economic, and demographic variation in neighborhoods. Second, there is a potential limitation in the contemporaneity of data used in this study, as TRO/VSO data were collected between 2016 and 2018 and the ACS survey data were collected from 2012 to 2016. Third, results from this study reflect associations at a specific moment in time when the analyses were conducted. Finally, although we used 12 neighborhood-level variables in our models, others could have been included. As a result, we are unable to speak to the weight of certain variables that were not included in analyses.

Despite these limitations, the novel analytic methods used here demonstrate that certain variables are more useful for explaining TRO and VSO density than others. Although results may be area dependent and unique to Virginia’s demographics at a specific moment in time, our results demonstrate that TROs and VSOs are likely to be located in neighborhoods and areas with greater socioeconomic disadvantage and possibly areas where there are more young adults, who are more likely to rent than own their homes [20]. Although density and placement of TROs and VSOs can impact the use of e-cigarettes and other tobacco products, so too can the marketing and point-of-sale practices of these retailers. Future research will not only need to investigate socioeconomic and demographic associations with TRO and VSO placement over time but also determine whether socioeconomic indices are associated with specific marketing (e.g., price promotions) and point-of-sale (e.g., age verification) practices.

## 5. Conclusions

Prior research has demonstrated the association between the proportion of minority and low-income residents and greater density of tobacco retail outlets. However, less is known about the degree to which these neighborhood-level characteristics are related to vape shop outlet density. Though, there is evidence to date suggesting that the same neighborhood-level characteristics reflecting socioeconomic disadvantage are associated with tobacco retail outlet density and vape shop outlet density. Our analyses of the spatial variation in tobacco retail outlets and vape shop outlets contributes to the existing literature by identifying differences in the spatial patterns and locations at high risk for the two types of retail outlets. Further, our results demonstrate that variables measuring socioeconomic disadvantage (e.g., higher renter occupied housing) were more important relative to variables measuring % minority population in explaining the association between neighborhood social disadvantage and tobacco retail outlet and vape shop outlet density. These results can be used to inform polices and target prevention efforts seeking to limit density in tobacco retail outlets and vape shop outlets in high-risk neighborhoods.

## Figures and Tables

**Figure 1 ijerph-17-02864-f001:**
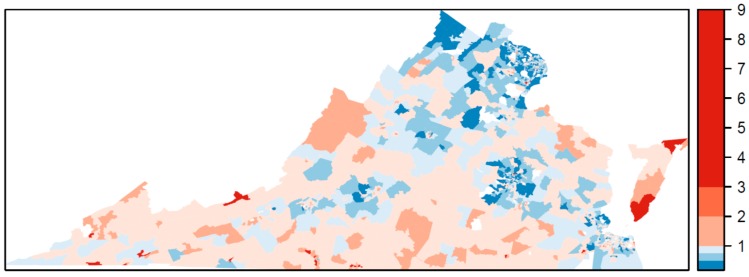
Estimated relative risk for tobacco retail outlets across Virginia census tracts, 2018.

**Figure 2 ijerph-17-02864-f002:**
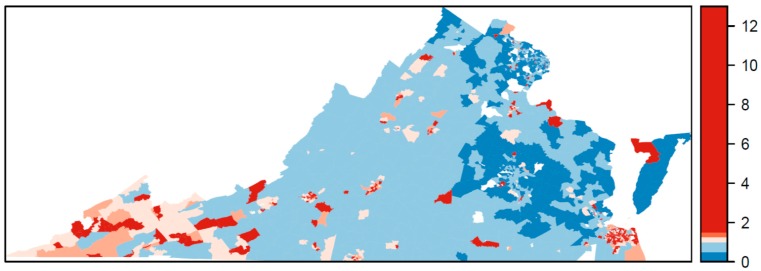
Estimated relative risk for vape shop outlets across Virginia census tracts, 2018.

**Figure 3 ijerph-17-02864-f003:**
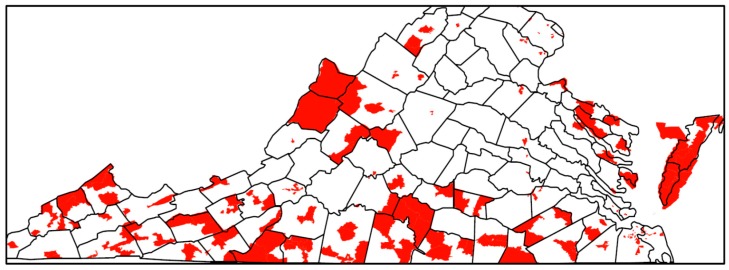
Census tracts with significantly elevated risk of tobacco retail outlets with county boundaries in Virginia, 2018.

**Figure 4 ijerph-17-02864-f004:**
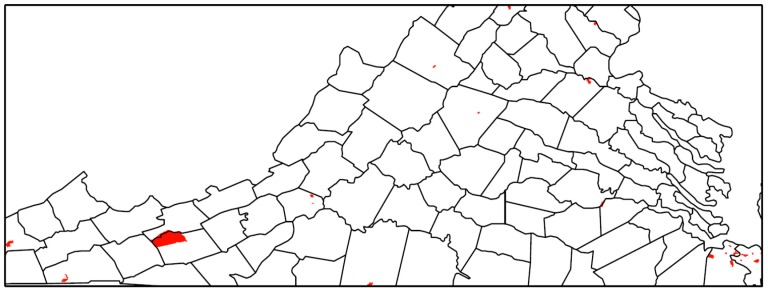
Census tracts with significantly elevated risk of vape shop outlets with county boundaries in Virginia, 2018.

**Table 1 ijerph-17-02864-t001:** Deviance information criterion (DIC) and effective number of parameters (pD) values for models explaining tobacco retail outlet and vape shop outlet rates in Virginia, 2018.

Model	Tobacco Retail Outlets		Vape Shop Outlets	
	DIC	pD	DIC	pD
1	8288.2	9.3	1121.2	4.3
2	7257.9	818.5	1032.9	150.7
3	7204.8	771.5	1018.5	140.1
4	6838.2	399.6	958.7	60.2

**Table 2 ijerph-17-02864-t002:** Empirically estimated weights of variables in the Neighborhood Disadvantage Index for tobacco retail outlets and vape shop outlets in Virginia, 2018.

Census Tract Variable	TRO Index Weight	VSO Index Weight
% Renter occupied housing units	0.23	0.46
Inverse median gross rent	0.20	0.04
% Without bachelor’s degree	0.10	0.03
Inverse median monthly housing costs	0.09	0.04
% Vacant housing units	0.09	0.04
Gini index of income inequality	0.07	0.04
% Hispanic population	0.07	0.20
Inverse median household income	0.05	0.03
% US citizen	0.04	0.03
% Families in poverty	0.03	0.04
% Households with public assistance income	0.03	0.03
% Black population	0.02	0.03

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
