# Peer review of "Neighborhood Disadvantage and Tobacco Retail Outlet and Vape Shop Outlet Rates"

_ijerph, 2020, doi:10.3390/ijerph17082864_

Round 1

Reviewer 1 Report

This is a timely study with implications in policy on vape and tobacco retailers. It identifies significant relationship between neighborhood level indicators and vape and tobacco retailer density. However, there are some major issues that have to be addressed.  The main issue has to do with a lack of a clear purpose of the study. Is this study a presentation of a new spatial model that outperforms more conventional tools, or is it the identification of clustering and important predictors of vape and tobacco retailer density? None of these two aims are adequately explored and developed in the paper. I propose focusing on either one and provide more evidence for the specific claims. Specifically:

  1. I am a bit confused about the purpose of the study. If the study proposes a model that better represents the data generating mechanism of spatial patterns compared to traditional models such as ANOVA or regression, as the authors suggest, then there should be an explicit comparison among the models.  These comparisons should be the main purpose of the paper, and discussion of the actual relationships identified by the ‘best’ model as secondary.  Actually the authors argue that the Bayesian method with correlated predictors “could help bring greater clarity to the ways in which TRO/VSO density varies in relation to neighborhood variables”.  However, they do not show that the new model is better in terms of clarity from the more parsimonious models of ANOVA or regression. The authors, could reformulate the purpose of the paper to identifying correlates of VSO as primary, since the results and discussion focus on that, rather on the superiority of their model compared to other models.
  2. One of the main arguments of the paper is that the Bayesian model proposed accounts for spatial correlation. The authors need to show that there is indeed spatial clustering of observations using an index, such as Moran’s I or something similar. If there is no spatial autocorrelation (and or correlation among variables) then the proposed model should not outperform ANOVA. 
  3. The TRO/VSO data were collected between 2016 and 2018, and the ACS survey data were collected from 2012-2016. That implies that some of the data from ACS (e.g. data from 2012) are linked with data of TRO/VSO from 2018.  Does this poses a limitation in the contemporaneity of the two data sources?
  4. The authors make the claim in the discussion that their results show “geographic variability in the placement of TRO and VSOs with statistically elevated risk areas (spatial clusters) 227 in the eastern portions, south central and western Appalachian part of the state.” Can they provide statistical support of this statement?

 Minor:

  1. I recommend replacing words that imply causality such as “influence” or ‘explaining’ with ‘associated’. For example: replace “important variables explaining TRO rates included” with “important variables associated with TRO rates included”.
  2. Make sure that all the elements of an equation are explained correctly. For example, on page 3 line 134, the letter ?I is not an element in the equation referred to. Also, not all symbols are explained (e.g. ?).
  3. I don’t think that 5000 burn-in iterations are sufficient for a complex model as the one proposed. I also, believe that the iterations should be increased significantly (from 30000 to > 60000), unless the authors can provide support that the estimation properties are appropriate for spatial models with ICAR random effects.

Author Response

Reviewer 1

This is a timely study with implications in policy on vape and tobacco retailers. It identifies significant relationship between neighborhood level indicators and vape and tobacco retailer density. However, there are some major issues that have to be addressed.  The main issue has to do with a lack of a clear purpose of the study. Is this study a presentation of a new spatial model that outperforms more conventional tools, or is it the identification of clustering and important predictors of vape and tobacco retailer density? None of these two aims are adequately explored and developed in the paper. I propose focusing on either one and provide more evidence for the specific claims.

RESPONSE: We appreciate and agree with Reviewer 1’s comment that this is a timely study with implications in policy on vape and tobacco retailers. However, Reviewer 1 also mentioned that there is a major issue with the manuscript, that being a lack of clear purpose for the study. To mitigate this concern, we have clarified in the text that “The objective of this study was to identify significant clusters and hotspots of vape and tobacco retailer density while finding important neighborhood socioeconomic status (SES) variables associated with outlet rates.” (Page 3, Lines 91-93). “To accomplish this objective, we developed Bayesian hierarchical models for TRO and VSO rates at the census tract level that handle spatial autocorrelation and correlated neighborhood variables in a neighborhood disadvantage index (NDI).” (Page 3, Lines 93-97).

Specifically:

  1. I am a bit confused about the purpose of the study. If the study proposes a model that better represents the data generating mechanism of spatial patterns compared to traditional models such as ANOVA or regression, as the authors suggest, then there should be an explicit comparison among the models.  These comparisons should be the main purpose of the paper, and discussion of the actual relationships identified by the ‘best’ model as secondary.  Actually the authors argue that the Bayesian method with correlated predictors “could help bring greater clarity to the ways in which TRO/VSO density varies in relation to neighborhood variables”.  However, they do not show that the new model is better in terms of clarity from the more parsimonious models of ANOVA or regression. The authors, could reformulate the purpose of the paper to identifying correlates of VSO as primary, since the results and discussion focus on that, rather on the superiority of their model compared to other models.

RESPONSE: In the revised manuscript, we have specified that the purpose of the paper is to model variation in vape and tobacco retailer density while identifying important neighborhood SES correlates. The mechanism for accomplishing this aim is through developing Bayesian hierarchical models that can handle correlated neighborhood SES variables and spatial autocorrelation in TRO and VSO rates at the census tract level. We have added three more parsimonious Bayesian models for explaining variation in outlet rates and demonstrate that the convolution mixture model (the original and most complex model) has the best model goodness-of-fit, as explained in further detail on Pages 4-6.

  1. One of the main arguments of the paper is that the Bayesian model proposed accounts for spatial correlation. The authors need to show that there is indeed spatial clustering of observations using an index, such as Moran’s or something similar. If there is no spatial autocorrelation (and or correlation among variables) then the proposed model should not outperform ANOVA. 

RESPONSE: In the revised analysis, we included a model (1) with exchangeable (independent) random effects at the census tract level as a comparison. The models (2-4) with spatial random effects have substantially better model goodness-of-fit than model 1 according to the deviance information criterion (DIC); see Table 1 on page 6. This demonstrates that there is substantial spatial correlation in the outlet rates that should be modeled.

  1. The TRO/VSO data were collected between 2016 and 2018, and the ACS survey data were collected from 2012-2016. That implies that some of the data from ACS (e.g. data from 2012) are linked with data of TRO/VSO from 2018.  Does this poses a limitation in the contemporaneity of the two data sources?

RESPONSE: We agree that the time of data collection for TRO/VSO data and ACS survey data poses a potential limitation in the contemporaneity of the two data sources. This has been added as a potential limitation of the existing research: “Second, there is a potential limitation in the contemporaneity of data used in this study as TRO/VSO data was collected between 2016 and 2018 and the ACS survey data was collected from 2012 to 2016” (Page 12, Lines 389-391).

  1. The authors make the claim in the discussion that their results show “geographic variability in the placement of TRO and VSOs with statistically elevated risk areas (spatial clusters) 227 in the eastern portions, south central and western Appalachian part of the state.” Can they provide statistical support of this statement?

RESPONSE: This comment refers to the statistically significant areas mapped in Figure 3 for TROs and Figure 4 for VSOs as defined using the exceedance probabilities described in the Statistical analysis section. We have added the direct reference to these figures for this statement. In addition, we have added the list of significant census tracts with county identifiers for TROs and VSOs in a supplemental materials file.

Minor:

  1. I recommend replacing words that imply causality such as “influence” or ‘explaining’ with ‘associated’. For example: replace “important variables explaining TRO rates included” with “important variables associated with TRO rates included”.

RESPONSE: We have made sure to replace all language in the manuscript that we consider to be causal.

  1. Make sure that all the elements of an equation are explained correctly. For example, on page 3 line 134, the letter ?is not an element in the equation referred to. Also, not all symbols are explained (e.g. ?).

RESPONSE: We have reviewed all elements of equations and have checked to make sure that they are explained correctly. We have corrected the error mentioned. The correct symbol for the random effect is .

  1. I don’t think that 5000 burn-in iterations are sufficient for a complex model as the one proposed. I also, believe that the iterations should be increased significantly (from 30000 to > 60000), unless the authors can provide support that the estimation properties are appropriate for spatial models with ICAR random effects.

RESPONSE: We have increased the number of MCMC iterations to 60,000 with a burn-in of 30,000 and checked that all parameters of inference have converged. These changes have also been made in the text on Pages 5-6.

Reviewer 2 Report

"Disadvantage of neighborhood and retail sale of tobacco Rates of stores and vape stores", is an interesting manuscript.

- We know from previous studies of state and local tobacco laws that the way a law is implemented may be the difference between an effective and ineffective policy. Authors should provide a context on tobacco control laws in Virginia, e.g. bans around school

- I suggest a comment from regarding: how does Bayesian estimates influence being an urban, suburban or rural population? And on the measure of the multiple tobacco retailers proximity, in the proposed model.

In Line 91: The hypothesis says that VSO rates should be more likely in areas with a higher percentage of White population. In Line 237: The second most important variable was % Hispanic population. I would appreciate a discussion about these ideas. I suggest a describe % smoking and vaping and population distribution of white, black and Latino or Hispanic by neighborhood in Virginia.

Author Response

Reviewer 2

"Disadvantage of neighborhood and retail sale of tobacco Rates of stores and vape stores", is an interesting manuscript.

RESPONSE: Thank you for the positive feedback.

- We know from previous studies of state and local tobacco laws that the way a law is implemented may be the difference between an effective and ineffective policy. Authors should provide a context on tobacco control laws in Virginia, e.g. bans around school

RESPONSE: To address this comment, we have provided a context on tobacco control laws in Virginia. Specifically, we have added the following to the text: “Within Virginia, there are no licensing requirements for the sale of tobacco and no existing regulations that affect where tobacco and nicotine containing products are old. Virginia also has one of the lowest excise taxes on both cigarette and non-cigarette products in the United States” (Page 3, Lines 123-125).

- I suggest a comment from regarding: how does Bayesian estimates influence being an urban, suburban or rural population? And on the measure of the multiple tobacco retailers proximity, in the proposed model.

RESPONSE: The Bayesian models are of the rate of TROs or VSOs per number of households in a census tract. Therefore, population density (i.e., urban, suburban, or rural) is directly considered in the models. The neighborhood SES variables also vary over space in relation to population density, so population density is also accounted for through covariate patterns. For retailers, the models directly consider the number of TROs or VSOs in a census tract. However, proximity between outlets within a census tract (e.g., average distance between retailers) is not considered in the models. These details have been added to Page 5, Lines 191-195.

In Line 91: The hypothesis says that VSO rates should be more likely in areas with a higher percentage of White population. In Line 237: The second most important variable was % Hispanic population. I would appreciate a discussion about these ideas. I suggest a describe % smoking and vaping and population distribution of white, black and Latino or Hispanic by neighborhood in Virginia.

RESPONSE: As suggested by Reviewer 2, we have added a discussion regarding the expectation that VSO rates should be more likely in areas with a higher % White population and our finding that % Hispanic population was an important variable. Specifically, we provide details on the growing population of Hispanic and Latinos within Virginia and the growing prevalence of tobacco and e-cigarette use within this group (Page 10, Lines, 332-347). Unfortunately, we are unable to describe % smoking and vaping by population subgroups for neighborhoods in Virginia as these data are not publicly available.

Reviewer 3 Report

The authors of this research studies the neighborhood disadvantage and tobacco retail outlet and vape shop outlet rates in Virginia. This study is interesting and well-written. Data were collected between 2016 and 2018. The use of Bayesian model is a bit surprising, but this method also makes this paper unique. I am providing a few suggestions here that I hope these comments can be helpful with authors’ revision.

  1. For a study like this, it is important to include a theoretical framework such as a social ecological model. If there are other frameworks available, the authors should consider them.
  2. The introduction should address the importance of studying Virginia. The current introduction section does not provide robust evidence of studying this state.
  3. Because the authors cover different topics in the introduction section, I would suggest the authors to create a few other subsections to differentiate the paragraphs.
  4. Can authors use simple language to describe the statistical analysis section? In its current form, it is difficult for readers, without sufficient background of Bayesian statistics, to understand the analysis.
  5. Any approvals from Institutional Review Board (IRB)?
  6. The statistical results should be presented with clear tables.

Author Response

Reviewer 3

The authors of this research studies the neighborhood disadvantage and tobacco retail outlet and vape shop outlet rates in Virginia. This study is interesting and well-written. Data were collected between 2016 and 2018. The use of Bayesian model is a bit surprising, but this method also makes this paper unique. I am providing a few suggestions here that I hope these comments can be helpful with authors’ revision.

RESPONSE: We appreciate the positive feedback that our study is interesting and well-written. We have addressed the suggestions below.

  1. For a study like this, it is important to include a theoretical framework such as a social ecological model. If there are other frameworks available, the authors should consider them.

RESPONSE: To address this comment, we have added a paragraph focused on the theoretical framework within the Introduction section of the manuscript. In this section, we explain that, “The social-ecological model provides a useful framework for examining the ways in which social-contextual factors (such race/ethnicity, income, housing) help to explain tobacco use, while also identifying areas of potential intervention at the individual, family/peer, community, and societal level…By identifying significant clusters and hotspots of vape and tobacco retailer density while finding important neighborhood SES variables that explain variation in TRO and VSO rates, we are able to identify communities that require intervention, while also determining patterns of social circumstance that occur with neighborhood social disadvantage that can be used to inform and improve future prevention and intervention efforts.” (Page 3, Lines 105-117).

  1. The introduction should address the importance of studying Virginia. The current introduction section does not provide robust evidence of studying this state.

RESPONSE: We agree that it is important to address the importance of studying Virginia. As the current introduction does not provide evidence for studying this state, we have included details under 2. Materials and Methods, Data Sources and Measures: “Virginia is an important state for the study of tobacco use… And although state rates for smoking are generally lower when compared to the national average, tobacco use remains the leading preventable cause of death within the state. Further, state-level data have demonstrated that tobacco use differs by geographic region, race/ethnicity, income, and education across the state – suggesting that these are important variables to investigate in our analyses” (Page 3, Lines 122-130).

  1. Because the authors cover different topics in the introduction section, I would suggest the authors to create a few other subsections to differentiate the paragraphs (e.g. Neighborhood characteristics and tobacco retail outlet density, Neighborhood characteristics and vape shop outlet density, Statistical approaches for examining associations between neighborhood characteristics and TRO/VSO density, and Research objectives).

RESPONSE: As suggested, we have created a few other subsections to differentiate paragraphs in the introduction section.

  1. Can authors use simple language to describe the statistical analysis section? In its current form, it is difficult for readers, without sufficient background of Bayesian statistics, to understand the analysis.

RESPONSE: In the revised manuscript, we have included additional models and descriptions of each model and model term in simple language on pages 4-5.

  1. Any approvals from Institutional Review Board (IRB)?

RESPONSE: Yes, this research was approved by the Virginia Commonwealth University Institutional Review Board. We have added these details to the revised manuscript: “Study protocols were approved by the Institutional Review Board at Virginia Commonwealth University” on Page 3, Lines 130-131.

  1. The statistical results should be presented with clear tables.

RESPONSE: We have added a table that shows the model goodness-of-fit and effective number of model parameters (Table 1 on page 6). The table of SES variable weights is now Table 2 (pages 9-10). We have also added a list of census tracts that are significantly elevated for TRO or VSO rates in a new supplemental material file.

Round 2

Reviewer 1 Report

Thank you for your detailed responses to my comments and suggestions.  I still have one request for clarification on an earlier comment:

I don't feel that may previous comment #4 was adequately addressed. My comment was referring to whether the authors have checked for spatial autocorrelation in the data. This is usually done by Moran's I, although there are other measures.  The use of the 'exceedance probabilities" does not qualify as an index whether adjacent areas (that is, those sharing a common boundary) had more similar area level residuals than would be expected under spatial randomness. That is what spatial autocorrelation refers to. Can the authors clarify if they checked for spatial clustering (autocorrelation) in the exploratory phase of their data analysis, and if yes how.  Moran's I is a very easy statistic to calculate. If the authors decide not to check for spatial autocorrelation then statements about geographic variability are not well supported.

Author Response

Thank you for your detailed responses to my comments and suggestions.  I still have one request for clarification on an earlier comment:

I don't feel that may previous comment #4 was adequately addressed. My comment was referring to whether the authors have checked for spatial autocorrelation in the data. This is usually done by Moran's I, although there are other measures.  The use of the 'exceedance probabilities" does not qualify as an index whether adjacent areas (that is, those sharing a common boundary) had more similar area level residuals than would be expected under spatial randomness. That is what spatial autocorrelation refers to. Can the authors clarify if they checked for spatial clustering (autocorrelation) in the exploratory phase of their data analysis, and if yes how.  Moran's I is a very easy statistic to calculate. If the authors decide not to check for spatial autocorrelation then statements about geographic variability are not well supported.

RESPONSE: We used a model selection strategy based on the goodness-of-fit, as measured by the DIC. This is described in the Methods and Results. The DIC results show that the models with spatial random effects have significantly better fit than models without spatial random effects. This indicates that there is significant spatial autocorrelation in the data. However, as requested we have added a test of spatial autocorrelation to the Results (second paragraph, line 251). The Moran’s I test is highly statistically significant (p=0.001) for TROs and marginally significant for VSOs (p=0.10).

Reviewer 3 Report

The authors have clearly addressed my comments and concerns. I believe the manuscript has reached a publishable rating. Thank you for your efforts.